# Structures of vesicular stomatitis virus glycoprotein G alone and bound to a neutralizing antibody

**Marie Minoves[1], Malika Ouldali[1]ʘ, Laura Belot[1]ʘ, Stéphane Roche[1], Sarah Johari[1], Magali Noiray[1], Eleftherios Zarkadas[2], Guy Schoehn[3], Yves Gaudin[1], Aurélie A. Albertini[1]***

**1** Institute for Integrative Biology of the Cell (I2BC), CEA, CNRS, Université Paris-Saclay, Gif-sur-Yvette, France, **2** Université Grenoble Alpes, CNRS, CEA, EMBL, ISBG, Grenoble, France, **3** Université Grenoble Alpes, CNRS, CEA, IBS, Grenoble, France

ʘ These authors contributed equally to this work.
* aurelie.albertini@i2bc.paris-saclay.fr

## Abstract

VSV G mediates viral entry *via* endocytosis. In the endosome, G undergoes a pH-dependent conformational change from pre- to post-fusion state, catalyzing membrane fusion. No complete structure of G has been reported so far. We present cryo-EM structures of G, isolated from virions using detergent, alone and in complex with the broadly neutralizing antibody 8G5F1 that binds all G conformations. The post-fusion structure reveals a novel rearrangement of the C-terminal part of the G ectodomain, showing that it undergoes a conformational rearrangement and stabilizes the post-fusion trimer by nesting into a groove between adjacent fusion domains. Structures of G-Fab complex show that the epitope belongs to a conserved antigenic site, explaining the broad neutralization capacity of the antibody. This work provides insights into the molecular basis of VSV G mediated fusion and antibody recognition, with potential implications for vaccine development, oncolytic virotherapy.

## Author summary

Vesicular stomatitis virus (VSV) enters host cells *via* endocytosis During this process, its surface glycoprotein G undergoes a low pH-triggered conformational change that drives fusion of the cellular and viral membranes. Although central to infection, a complete high-resolution structure of VSV G has long been missing. Using cryo-electron microscopy, we determined the structure of purified VSV G, both alone and in complex with a broadly neutralizing antibody that recognizes several conformations of VSV G. Our results reveal that in the post-fusion state, the C-terminal part of the ectodomain folds back and inserts between neighboring protomers, stabilizing the trimeric post-fusion assembly. The antibody-bound

**Data availability statement:** The map was deposited in the electron microscopy data bank (EMDB) with ID EMD-52731, EMD-52152, EMD-52169, EMD-52174 and the atomic model in the protein data bank (PDB) with ID 9I8Q, 9HGN, 9HH9, 9HHR for the pre-fusion conformation, the post-fusion conformation, the pre-fusion conformation in complex with 8G5F11 Fab and the post-fusion conformation in complex with 8G5F11 Fab, respectively. All data supporting the findings of this study are within the article and its Supplementary Information.

**Funding:** This work was supported by a grant from the Agence Nationale de la Recherche, France, (ANR-22-CE11-009-01, Grly) attributed to A.A.A., and a grant from the Fondation pour la Recherche Médicale, France, (EQU202103012746) to Y.G. This work also benefited from the I2BC platforms, which are supported by the French Infrastructure for Integrated Structural Biology (ANR-10-INSB-05-05). This work also used the platforms of the Grenoble Instruct-ERIC center (ISBG; UAR 3518 CNRS-CEA-UGA-EMBL) within the Grenoble Partnership for Structural Biology (PSB), supported by FRISBI (ANR-10-INBS-0005-02) and GRAL. The IBS/ISBG electron microscope facility is supported by the Auvergne-Rhône-Alpes Region, the Fondation pour la Recherche Médicale, the fonds FEDER, and the GIS-Infrastructures en Biologie Santé et Agronomie (IBISA). The funders had no role in the design of the study, the collection, analysis or interpretation of the data, the decision to publish, or the preparation of the manuscript.

**Competing interests:** The authors have declared that no competing interests exist.

structures further identify a conserved epitope that remains accessible across conformations, explaining how it can neutralize vesiculoviruses so broadly.

Together, these findings expand our understanding of how VSV G mediates membrane fusion and how antibodies can block it, offering structural clues that could guide the design of vaccines and oncolytic vectors.

## Introduction

Vesicular stomatitis virus (VSV) is the prototype of the Vesiculovirus genus in the Rhabdoviridae family. It is an enveloped, bullet-shaped virus capable of infecting a wide range of hosts, including mammals and insects, and is responsible for epizootic outbreaks in cattle in North America [1]. Its negative-strand RNA genome encodes five structural proteins among which the glycoprotein (VSV G) is the sole transmembrane protein. VSV G plays a crucial role in initiating the viral cycle. It is responsible for both receptor recognition and membrane fusion. Interaction of VSV G with a cellular receptor, particularly members of the LDL-R family [2,3], triggers clathrin-mediated endocytosis of VSV [4,5]. Then, within the acidic environment of the endosome, VSV G catalyzes fusion between viral and endosomal membranes. Indeed, VSV G undergoes a low pH-induced fusogenic conformational change from a pre- toward a post-fusion state. This structural transition results in the transient exposure of hydrophobic motifs (called fusion loops) capable of interacting with the target membrane and destabilizing it, ultimately leading to its fusion with the viral envelope. A particularity of VSV G, shared with other rhabdovirus glycoproteins, is that there is a pH-dependent equilibrium between the pre-fusion state [6–8], the post-fusion state, and intermediate conformations on the conformational change pathway. Consequently, the structural transition induced by low pH is reversible [9–11], which is not the case for other viral fusogens whose pre-fusion conformation is metastable [12,13].

VSV G is a type I transmembrane protein. After cleavage of the amino terminal signal peptide, the mature glycoprotein is 495 amino acids (aa) long, the major part being its 446 residues long ectodomain ($G_{ect}$). To date, the three-dimensional (3D) structures of VSV G soluble ectodomains have been determined using X-ray crystallography in both the pre-fusion [14,15] and post-fusion states [16]. These soluble ectodomains were obtained either by proteolytic cleavage at the viral surface or through recombinant expression, with the membrane anchor removed in all cases [8]. Crystal structures of VSV $G_{ect}$ are available in both pre-fusion [14,15] and post-fusion trimeric conformations [16]. These structures revealed that VSV G shares a common fold with other viral fusion glycoproteins (including gB of herpesvirus [17] and gp64 of baculoviruses [18]), which defines class III membrane fusion proteins [19,20]. Subsequently, crystal structures of Chandipura virus glycoprotein (CHAV G, another glycoprotein of the Vesiculovirus genus) soluble ectodomain were also solved, corresponding to the post-fusion trimeric conformation [21] and two monomeric intermediates [22]. More recently, the structure of VSV $G_{ect}$ in complex with its cellular receptor binding

domains (*i.e.,* CR2 and CR3 cysteine-rich domains of the low-density lipoprotein receptor (LDL-R)) was resolved, revealing the organization of the receptor-binding site on VSV G [2].

$G_{ect}$ includes three distinct structural domains (nomenclature defined in Fig 1A): the fusion domain (FD), the trimerization domain (TrD), and the pleckstrin homology domain (PHD) that keep their tertiary structure during the conformational change. These domains are connected by hinge regions (named R1, R2, R3, R4 and R5) that refold during the structural transition, and therefore reposition structural domains [23,24]. Finally, the C-terminal domain (CTD) connects the ectodomain to the transmembrane domain (TMD) (Fig 1A).

To date, no complete structure (*i.e.* including the TMD and the intra-viral or cytoplasmic domain -ID-) of VSV G has been reported. It should also be noted that only part of the structure of the CTD is known (up to residue 432 for the pre-fusion state and residue 410 for the post-fusion state). Interestingly, several mutagenesis studies suggest that the CTD, along with the TMD, are critical for G to mediate membrane fusion, especially in the final steps of this process [25–28]. Thus, the mechanistic steps of G refolding and membrane fusion process are not fully understood due to the lack of structural information on the CTD and TMD.

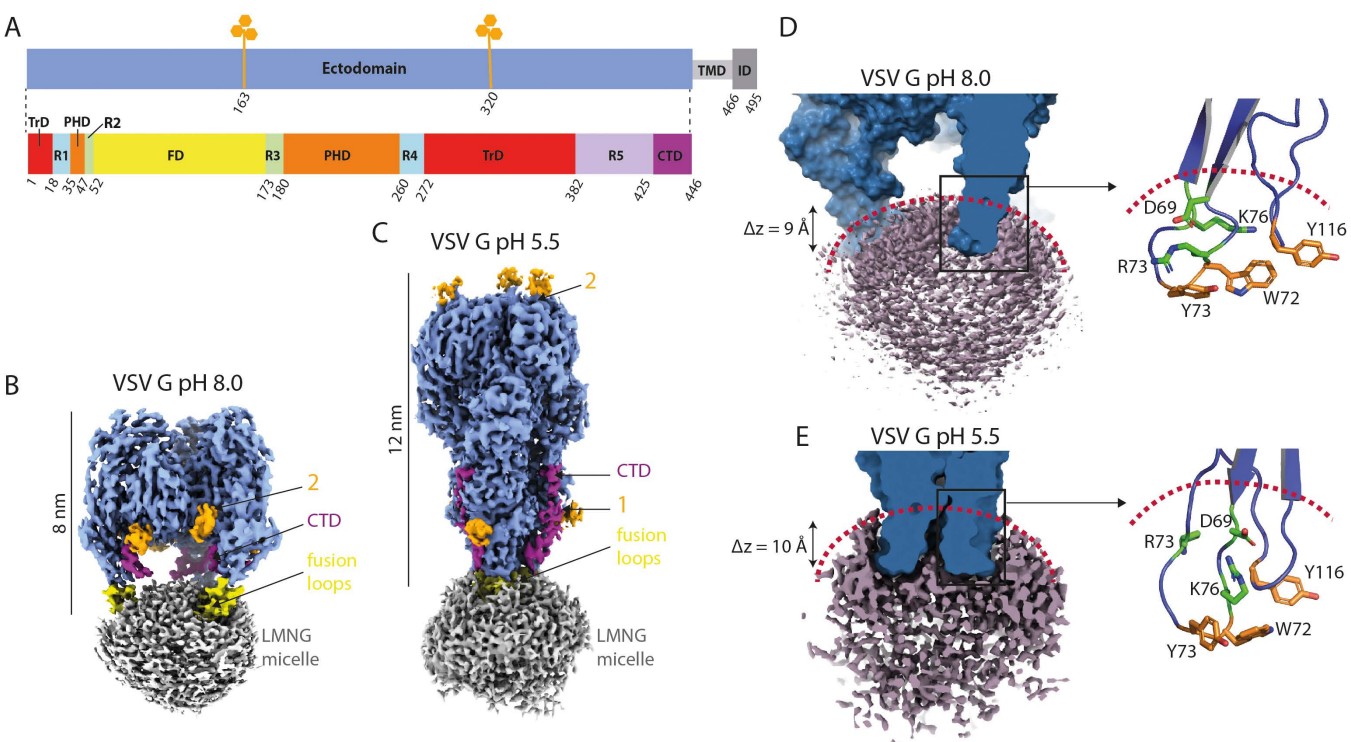

**Fig 1. VSV G cryo-EM structures in its pre- and post-conformations.** (A) Upper panel: Schematic diagram of VSV G ectodomain (in sky blue) with the position of the two glycosylation sites (N163 and N320) indicated in orange. The transmembrane domain (TMD) and intraviral domain (ID), which are not resolved in our structures, are represented in shades of grey. Lower panel: Domain organization of VSV G ectodomain. The trimerization domain (TrD) is shown in red, the pleckstrin homology domain (PHD) in orange, and the fusion domain (FD) in yellow. Domains are connected by five segments (named R1 to R5) that refold during conformational change: R1 and R4 are in cyan, R2 and R3 are in green, R5 and the C-terminal domain (CTD) are shown in light and deep purple, respectively. (B-C) Cryo-EM density map of full-length VSV G solubilized in LMNG and incubated at pH 8.0 (B) and at pH 5.5 (C). The G ectodomain is depicted in sky blue, the LMNG micelle in grey, and the CTD in magenta. The part of the fusion domain inserted in the micelle is colored in yellow. The starting residues of the N-linked glycosylation chains are depicted in orange, with the positions of N163 and N320 indicated by labels 1 and 2, respectively. (D-E) Close-up view of the fusion loops insertion into the micelle at pH 8.0 (D) and pH 5.5 (E). The outline of the micelle is indicated by the red dotted line. The hydrophilic residues located in the hydrophilic shell are depicted in green. The hydrophobic residues in the fusion loops are in orange.

VSV G ectodomain is the primary target of neutralizing antibodies. Although the antigenicity of VSV G has been characterized [29,30], no structural data of any VSV G-monoclonal antibody (mAb) complex have been reported so far.

Here, we report a cryo-electron microscopy (cryo-EM) study of detergent-solubilized full-length VSV G purified from viral particle. This study allows us to complete the structures of the ectodomain in its pre- and post-fusion conformations in a more physiological context. In particular, we extend our knowledge of the structure of the CTD in the post-fusion conformation and provide data that indicate that the TMDs are not visible in the detergent micelle, suggesting that they are rather mobile within the membrane. We also describe the first structures of VSV G bound to a commercial neutralizing antibody (mAb 8G5F11). This mAb binds G in both its pre- and post-fusion conformations. These first structures of a complex between G and a neutralizing antibody allowed the characterization of the epitope in detail and the identification of key residues on VSV G involved in this interaction. Globally, this work increases our knowledge of the structure of VSV G, the most widely used viral glycoprotein for the delivery of cargo and in gene therapy for lentivirus pseudotyping, and its low-pH induced refolding pathway.

## Results

### Overall structures of detergent solubilized VSV G in the pre- and post-fusion states

To gain insight into the full-length structure of VSV G, i.e., the ectodomain plus the TMD and ID (Fig 1A), we developed a protocol to isolate VSV G (Indiana strain) from concentrated virus particle preparations without any affinity purification tag. Briefly, VSV G was extracted directly from viral membranes using lauryl maltose neopentyl glycol (LMNG) and subsequently purified to homogeneity through successive chromatography steps (S1 Fig). We then performed cryo-EM single-particle analysis (SPA) on VSV G incubated at pH 8.0 (VSV G pH 8.0) and at pH 5.5 (VSV G pH 5.5) (S2, S3 Figs, S1 Table).

2D class averages were generated for both VSV G pH 8.0 and VSV G pH 5.5. Analysis of the 2D classes revealed a preferential orientation of the molecule in both conditions, with approximately 80% of the particle exhibiting top views (S2-S3 Figs). Using automated particle picking to enrich the data set with sides views, data processing yielded reconstructions with an overall resolution of 2.9 Å for both the high- and low-pH structures (S1B- S1C Fig).

The overall structure of the ectodomain in both reconstructions was consistent with previous studies, showing a trimeric spike with dimensions in height of 8 nm at pH 8.0 and 12 nm at pH 5.5, corresponding to the pre- and post-fusion conformations [8,31], respectively (Fig 1B-1C) and associated with a detergent micelle at the FD tips. Notably, the presence in the sample of TMD and ID did not appear to influence the global structure of VSV G ectodomain (Fig 1B-1C). Indeed, structural overlay of the crystallographic and cryo-EM derived models yielded root-mean-square deviations (RMSD) for the Cα backbone of 0.78 Å for the pre-fusion structure and 0.69 Å for the post-fusion structure (PDB code 6TIT [15] and 5I2M [16] respectively) (S2 Table). Additionally, there was a very clear density for the starting residues of the two N-linked glycosylation chains (attached to N163 and N320) [32] in both cryo-EM structures (Fig 1B-1C). It is worth noting that comparison with the crystallographic structure of the post-fusion state of VSV G [16] revealed an additional density corresponding to part of the CTD in the ectodomain of the post-fusion cryo-EM structure (Fig 1C, in magenta).

Despite repeated attempts to visualize the detergent micelle content using several computational cryo-EM methods (such as 3D-classification, 3DVA, 3DFLEX and several masking strategies), TMDs (resp. the IDs) were not discernible within (resp. in the vicinity of) the micelle (Fig 1B-1C) in both conformations, although SDS-PAGE did not reveal any proteolysis (S1 Fig).

On the other hand, the tips of FDs were visible inside the micelles, suggesting that the absence of detection of TMDs in the density is due to the fact that TMDs are mobile and do not have a fixed position within the micelle. Measurements indicate that the penetration depth of FDs into the micelle is approximately 9 Å for both conformations (Fig 1D-1E). This is consistent with the organization of the micelle of LMNG having a hydrophobic core radius of 14 Å and a hydrophilic shell

thickness of 7 Å [33]. This allows the accommodation of the hydrophobic residues in the fusion loop in the hydrophobic core as well as several polar or charged residues in the hydrophilic shell (Fig 1D-1E).

## Reorganization of the C-terminal linker region

The resolution of the CTD in the post-fusion conformation is approximately 4.5 Å, allowing for its partial tracing up to residue 426 (Fig 2A). For each protomer, the traced segment extends into a predominantly hydrophilic groove (Fig 2B) located between this protomer (protomer A) and the neighboring one (protomer B). Structural analysis reveals a hydrophobic patch on the CTD, composed of residues L423 and F425, which interact closely with residue I82 located on the FD of the same protomer (protomer A), as well as residues L105 and P107 of the FD of the neighboring protomer (protomer B) (Fig 2C and S4 Fig). These interactions, which are mainly hydrophobic with very few polar contacts (S4 Fig), stabilizes

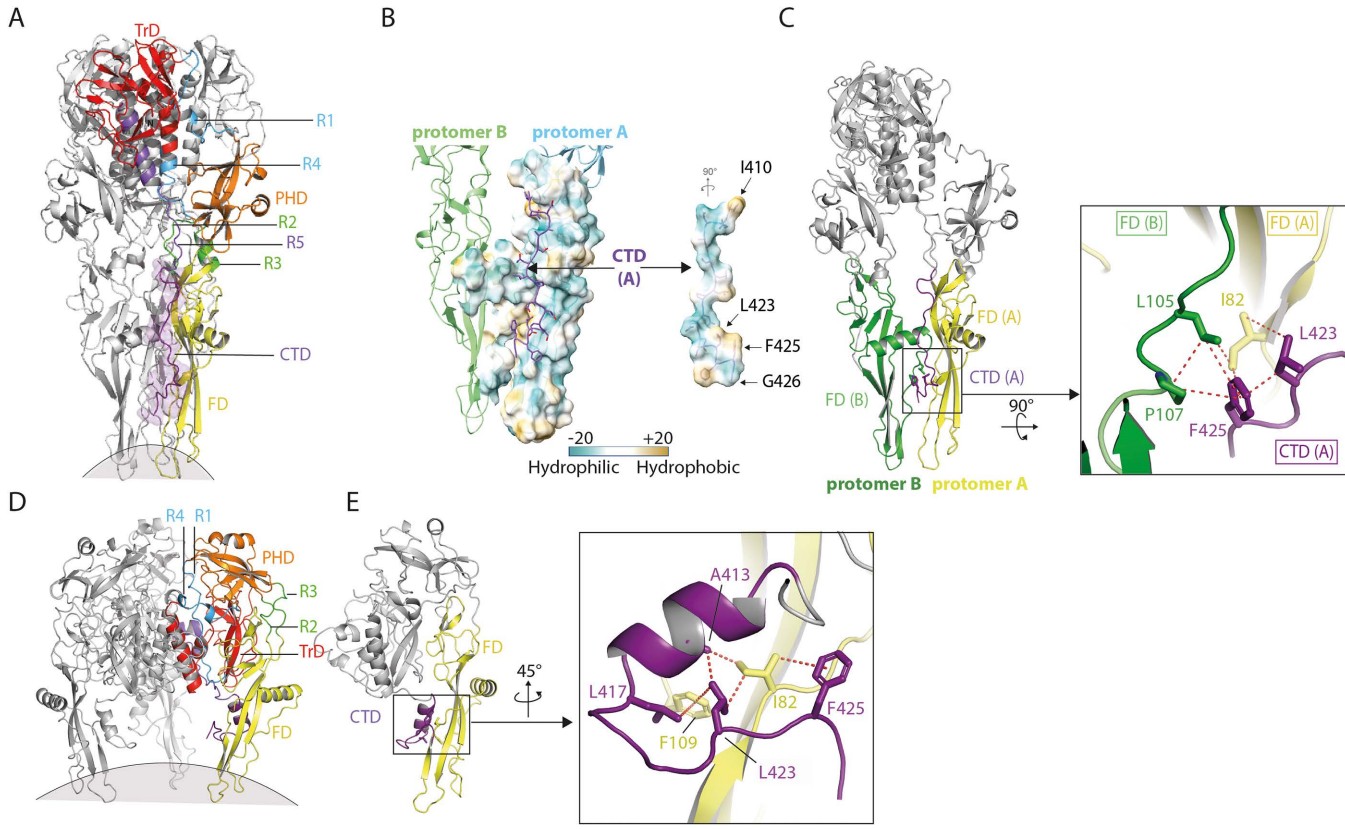

**Fig 2. VSV G C-terminal domain (CTD) rearrangement.** (A) Ribbon representation of VSV G trimer in its post-fusion conformation, fitted and traced up to residue 426 in the cryo-EM map obtained at pH 5.5. The same color code as in Fig 1A is used. (B) Surface hydrophobicity representation of the interaction between segment 410-426 of chain A (right) and neighboring FD (left) in the post-fusion conformation. The hydrophobicity scale is indicated. (C) Hydrophobic patch on the CTD-FD interface in the post-fusion conformation; Left panel: Overall view of VSV G trimer showing FD from protomer A (in yellow) and FD from protomer B (in green) with the CTD from protomer A in magenta. Residues displayed as sticks contribute to the stabilizing of the CTD-FD interface in the post-fusion state. Right panel: close-up on the hydrophobic patch formed at the level of the CTD (involving L423 and F425) interacting with I82 from the FD of protomer A (in yellow), and L105 and P107 from the FD of protomer B (in green). These residues stabilize the CTD-FD interface in the post-fusion conformation. Putative hydrophobic interactions are indicated by red dashed lines. (D) Ribbon representation of VSV G trimer in its pre-fusion conformation, fitted and traced up to residue 427 in the cryo-EM map obtained at pH 8.0. The same color code as in Fig 1A is used. (E) Close-up view of VSV G CTD in the pre-fusion conformation. The inset shows the folding of the CTD and a detailed view of the environment surrounding I82, L423 and F425 in VSV G pre-fusion conformation. Putative hydrophobic interactions are indicated by red dashed lines.

the C-terminal domain against the FD. Sequence alignment of Vesiculovirus glycoproteins indicates that the hydrophobic nature of these residues is conserved among the genus (S5 Fig).

In the pre-fusion conformation, the overall structure of the ectodomain was similar to previously determined crystal structures [15], with the CTD modeled up to residue 427 (Fig 2D-2E). In both X-ray crystallography and cryo-EM pre-fusion structures, the CTD forms a small helix (res 409 to 416) followed by a β-turn between P418 and E421 and a short β-hairpin (res 422 to 432, only visible up to res 427 in the cryo-EM structure) forming a hook-like structure (Fig 2E). This indicates that the presence (or absence) of the TMD of VSV G does not interfere with its fold. The C-terminal part of VSV G is further stabilized by the insertion of F109 into a hydrophobic pocket composed of residues I82, A413, L417, and L423, which position the CTD in close proximity to the FD. Notably, as in the post-fusion conformation, it is also the hydrophobic residues L423 and F425 on the CTD that form a hydrophobic cluster with I82 on the FD (Fig 2E). Thus, this hydrophobic cluster is present in both the pre- and post-fusion states and, in both cases, it locks the CTD in close proximity to the FD (Fig 2C and 2D-2E). However, the organization of the CTD is completely different between the two structures although mainly involving hydrophobic interactions (Figs 2C, 2D-2E and S4). Thus, the CTD, as so R1, R2, R3, R4 and R5 segments, refolds during the low-pH induced structural transition of VSV G. In fact, the CTD and R5 are part of a single long segment that refolds during G conformational change.

## Neutralization of VSV by mAb 8G5F11

The commercial monoclonal antibody 8G5F11 [29] neutralizes VSV and other Vesiculoviruses [30]. To characterize the interaction of VSV G with mAb 8G5F11, we incubated purified mAb at various pH values with magnetic beads coated with protein A, followed by the addition of purified full-length VSV G pre-incubated at pH values between 5.0 and 8.0. After an incubation of 30 minutes, beads were washed, and the associated proteins were analyzed by SDS-PAGE. These interaction experiments demonstrated that 8G5F11 is able to bind VSV G over a range of pH values from 8.0 to 5.0 (Fig 3A). This suggested that 8G5F11 recognizes several VSV G conformations.

Additionally, we employed bio-layer interferometry (BLI) to investigate the binding parameters of coated mAb 8G5F11 to VSV $G_{ect}$ at pH 8.0. For this, we used the thermolysin-generated VSV $G_{ect}$ (aa residues 1–422) that is monomeric in solution [8]. Analysis of the BLI data revealed that mAb 8G5F11 exhibited a dissociation constant in the nanomolar range for VSV $G_{ect}$ at pH 8.0 (Fig 3B, S3 Table).

We engineered the corresponding Fab fragment of mAb 8G5F11. For that, variable heavy (VH) and light (VL) chains of mAb 8G5F11 were sequenced and cloned into an insect cell expression vector (S6 Fig). The construction resulted in a Fab C-terminally fused to a strep-TagII. After establishment of the stable S2 cell line, recombinant Fab was purified from cell culture media by affinity chromatography followed by SEC (S6 Fig). Interaction experiments conducted under the same conditions as for the mAb 8G5F11 (except that the magnetic beads were coated with strep-tactin) show that Fab 8G5F11 binds VSV G across the same pH range, with BLI measurements also indicating a binding affinity in the nanomolar range (Fig 3C-3D).

To determine the $IC_{50}$ values of mAb 8G5F11 and its corresponding Fab, we performed neutralization experiments by pre-incubating a recombinant VSV that express eGFP (VSV-eGFP) with serial dilutions of mAb or Fab. After 30 minutes of incubation, the inoculum was transferred onto HEK-293T cells. After 5 hours of infection, the percentage of infected cells (expressing eGFP) was measured by flow cytometry. Both mAb 8G5F11 and Fab 8G5F11 inhibit VSV infection in a dose-dependent manner. Calculated viral titers were plotted against the concentration of mAb or Fab and the dose-response curve was fitted using the Hill equation (Fig 3E). The estimated $IC_{50}$ value for the Fab 8G5F11 was approximately 40 nM, whereas it decreased to 0.6 nM when neutralization was performed with the mAb 8G5F11. Fab 8G5F11 neutralization curve exhibited a Hill coefficient of 1.2 ± 0.17 whereas mAb neutralization curve had a Hill coefficient of 3.35 ± 0.79 indicating some cooperativity effect (Fig 3E). This is consistent with the divalent nature of the mAb, which

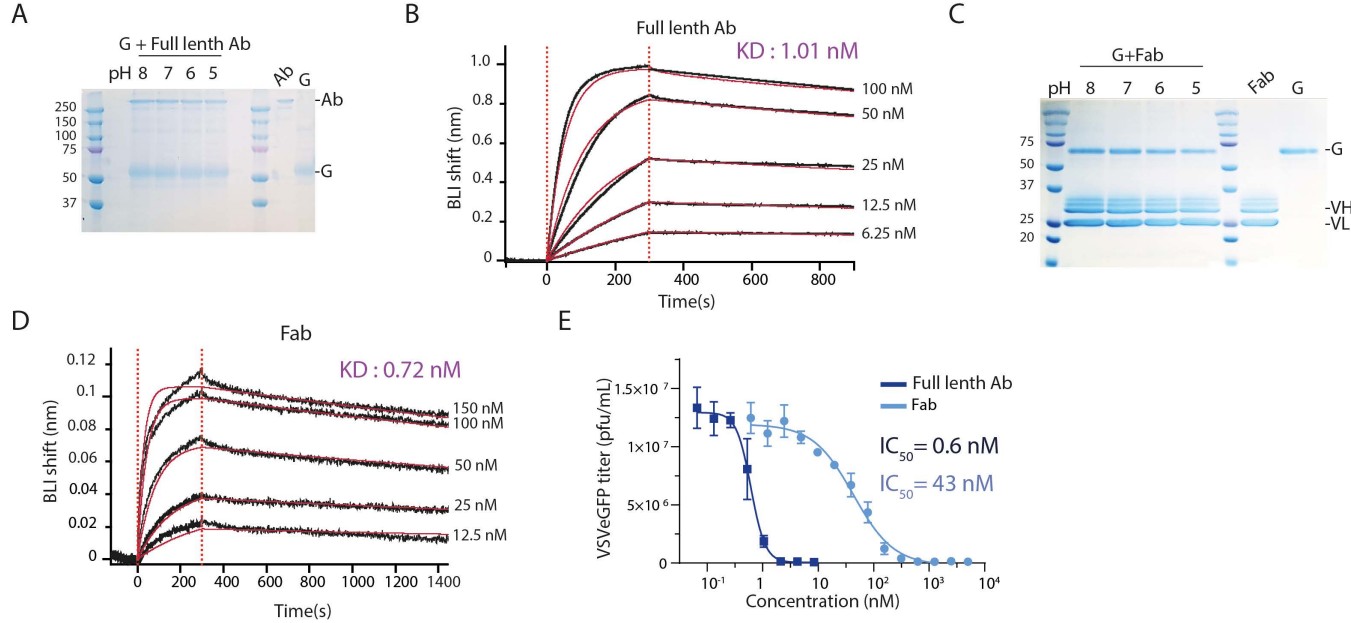

**Fig 3. mAb 8G5F11 interacts with the pre- and the post-fusion conformations of VSV G.** (A) Coomassie-stained SDS-PAGE analysis (under non-reducing conditions) of interaction experiments between VSV G and mAb 8G5F11 bound to protein A coated magnetic bead, incubated at various pHs. (B) BLI sensograms showing the binding kinetics of VSV $G_{ect}$ to mAb 8G5F11 at pH 8.0. Experimental curves (black) were fitted (red) using a 1:1 binding model. The calculated KD is indicated on the panel. (C) Coomassie-stained SDS-PAGE analysis (under reducing conditions) of interaction experiment between VSV G and Fab 8G5F11 bound to strep-Tactin coated magnetic beads, incubated at various pH values. (D) BLI sensograms showing the binding kinetics of VSV $G_{ect}$ to 8G5F11 Fab at pH 8.0. Experimental curves were fitted using a 1:1 binding model. The calculated KD is indicated on the panel. (E) Neutralization curve of VSVe-GFP by mAb (dark blue) and Fab (light blue). VSV-eGFP was preincubated with increasing concentrations of mAb or Fab. At 5 hours *p.i.,* the percentage of infected cells determined by counting the number of cells expressing eGFP using flow cytometry. This was used to calculate the infectious viral titer. Data depict the mean with standard error from experiments performed in triplicate. Average $IC_{50}$ value are indicated.

induces avidity-enhanced binding to VSV G, as well as with its larger size, which increases steric hindrance, potentially decreasing the number of mAbs required to neutralize a virion compared to that of Fabs.

## Structural characterization of Fab 8G5F11 bond to VSV G

To better understand how mAb 8G5F11 binds VSV G at different pH values, we performed cryo-EM SPA on VSV G in complex with Fab 8G5F11 at pH 8.0 and pH 5.5 (S6-S8 Figs, S1 Table). The cryo-EM structure of VSV G in complex with the Fab revealed a stoichiometric binding of one Fab per protomer at both pHs (Fig 4A and 4C). At pH 8.0, the Fab binds each VSV G protomer at the top of the molecule (Fig 4A-4B) with an angle of approximately 45° with respect to the axis 3 of the trimer (Fig 4B). At pH 5.5 the binding site is the same but the relative orientation of the Fab is different with an angle of 135° (Fig 4D).

The binding of the Fab did not influence the structure of the ectodomain; at pH 8.0 VSV G is in its pre-fusion conformation and at pH 5.5 VSV G is in its post-fusion conformation. The complexes between G and the Fab did not allow us to determine the structure of new parts of G.

In both structures, the interface between the Fab and VSV G is the same and principally involves the variable heavy chain (VH) of the Fab binding to the VSV G PHD (Fig 4B, 4D–4F). The interaction is mainly hydrophilic. Three VSV G residues (D241, K242, and D243) are located at the core of the interface between G and 8G5F11 paratope (Fig 4E). The carboxylic group of D241 establishes polar contact *via* its side chain with residue H53 of the Fab (Fig 4F). The side chain of

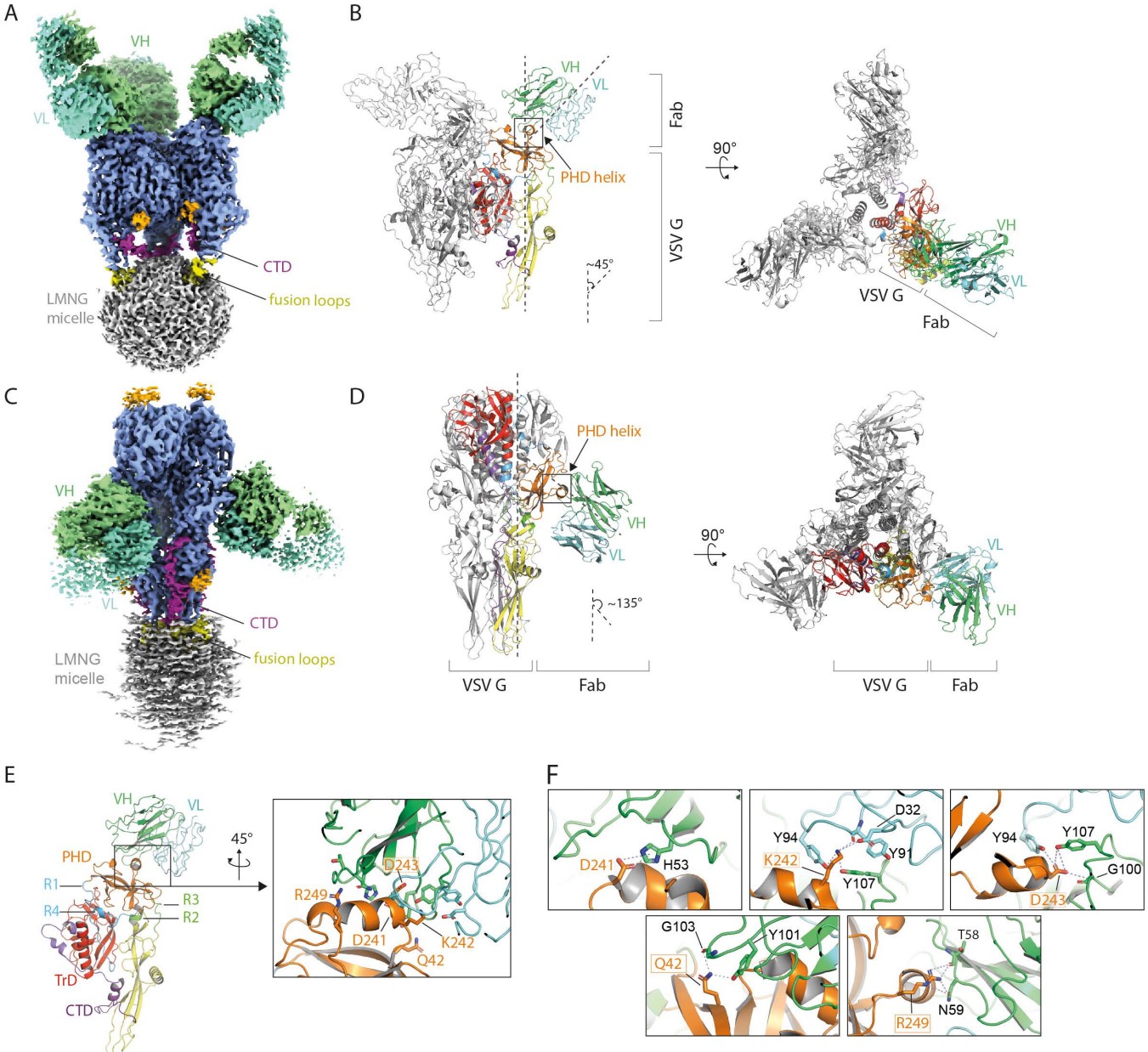

**Fig 4. Structural basis for the neutralization of VSV by mAb 8G5F11.** (A) Cryo-EM density map of VSV G in complex with Fab 8G5F11 at pH 8.0 shown in side view orientation. One 8G5F11 Fab molecule was observed to bind to each protomer of VSV G. G ectodomain is in sky-blue, the LMNG micelle in grey, the starting residues of the glycosylation chains are in orange. Fusion loops are in yellow and CTD in magenta. Fab variable heavy (VH) and variable light (VL) chains are in green and in cyan respectively. (B) Ribbon diagram of VSV G trimer in the pre-fusion conformation in complex with Fab 8G5F11, viewed from the side (left panel) and from the top (right panel). The VSV G model was fitted in the density and traced up to residue 426. The VH and VL chains of the 8G5F11 Fab were modelized and fitted in the identified density. The same color code as in Fig 1A is used for VSV G. Fab VH chain is in green and VL chain is in cyan. (C) Cryo-EM density map of VSV G in complex with Fab 8G5F11 at pH 5.5, shown in side orientation. The same color code as in Fig 4A is used. (D) Ribbon diagram of VSV G post-fusion trimer in complex with Fab 8G5F11, viewed from the side (left panel) and from the top (right panel). As in the pre-fusion state, one molecule of Fab binds each VSV G protomer of the post fusion trimer. The VSV G model was fitted in the density and traced up to residue 426, while the VH and VL chains of the 8G5F11 Fab were fitted in the identified density. The same code for G as in Fig 1A is used. Fab VH chain is in green and VL chain is in cyan. (E) Interaction between 8G5F11 and VSV G. Key residues are depicted in stick representation and are highlighted in the boxed panel. For clarity only, labels of key residues on VSV G are indicated. (F) Close up on the residues involved in the antibody-antigen interaction at high pH. VSV G PHD is in orange, and the key residues interacting with 8G5F11 are depicted in sticks.

D243 establishes a cluster of polar contacts with the hydroxyl groups of Y94 (on the VH chain) and Y107 (on the VL chain) as well as with the carbonyl of the main chain of G100 (on the VH chain) (Fig 4F). VSV G K242 ε-amino group interacts with the hydroxyl group of Y94 carboxyl group of D32 and the main chain of Y91 (on the VL chain) with the hydroxyl group of Y107 (on the VH chain) (Fig 4F). Two additional residues on VSV G (Q42 and R249) located outside of the core also contribute to the interaction. The amide group of Q42 side chain establishes hydrogen bonds with the carbonyl of the main chain of G103 and the hydroxyl group of Y101 on the VH chain of the Fab. Finally, the guanidinium group of R249 on VSV G establishes H-bonding with N59 side chain and the T58 main chain (both on the VH chain of the Fab) (Fig 4F).

## Characterization of key residues for 8G5F11 binding

To investigate the contribution of residues constituting the mAb 8G5F11 binding site on VSV G (Fig 4E), we generated alanine mutants at positions Q42, D241, K242, D243, R249 of G. HEK-293T cells were transfected with plasmids encoding either wild type (WT) VSV G or VSV G alanine mutants ($G_{mut}$: Q42A, D241A, K242A, D243A or R249A). Surface expression of the VSV G mutants was confirmed using a conformational probe, GST-CR3 (glutathione S-transferase fusion with the CR3 domain of the LDL-R [2]), labeled with a fluorescent dye ATTO$^{550}$ (Fig 5A). Indeed, GST-CR3 binds VSV G in its pre-fusion conformation and its binding site [2,15] is distinct from the binding site of mAb 8G5F11 (S9 Fig). Flow cytometry analysis confirmed that all mutants were correctly expressed at the cell surface, as evidenced by their efficient binding to the CR3-GST conformational probe (Fig 5B). Then, we assessed the ability of VSV G mutants to bind mAb 8G5F11 and its corresponding Fab on the cell surface. Mutations D241A and D243A induced a significant reduction in binding for both the mAb and Fab 8G5F11 (Fig 5C). Afterwards, we evaluated whether these mutant glycoproteins (VSV $G_{D241A}$ and VSV $G_{D243A}$) could maintain viral infection. We employed a recombinant VSV (VSVΔG/G) lacking the G envelope gene, replaced by the green fluorescent protein (GFP) gene pseudotyped with VSV glycoprotein (Fig 5D). The incorporation of VSV G mutants into the membrane of VSV pseudotypes was verified by Western-blot analysis (Fig 5E). Indeed, the VSVΔG/$G_{D241A}$ and VSVΔG/$G_{D243A}$ pseudotypes exhibit an equivalent level of glycoprotein incorporation to that observed with VSVΔG/$G_{WT}$. Infectivity of VSV pseudotypes in HEK-293T cells was determined by flow cytometry, measuring the percentage of infected cells 16 hours *p.i.* In the absence of antibody, both VSVΔG/$G_{WT}$ and VSVΔG/$G_{mut}$ pseudotypes exhibited comparable infectious titers, approximately $2 \cdot 10^7$ pfu/ml (Fig 5F, left panel). We then performed neutralization experiments by pre-incubating VSVΔG/$G_{WT}$ and VSVΔG/$G_{mut}$ inoculum with serial dilutions of mAb or Fab 8G5F11 (Fig 5F). As expected, VSVΔG/$G_{WT}$ pseudotype was efficiently neutralized by mAb and Fab 8G5F11, with $IC_{50}$ values of 0.2 nM and 10 nM respectively (Fig 5F, right panel). In contrast, VSVΔG/$G_{D241A}$ and VSVΔG/$G_{D243A}$ pseudotypes were not neutralized by Fab or mAb 8G5F11 at either concentration, indicating that these residues play an important contribution to mAb 8G5F11 binding and neutralization activity (Fig 5F, left panel). It is noteworthy that the acidic character of the residues corresponding to D241 and D243, is widely conserved among vesiculoviruses (S5 Fig). This explains why 8G5F11 is broadly neutralizing [30].

## 8G5F11 binding interferes with membrane fusion

As mentioned above, the binding site of 8G5F11 is distinct from that of the receptor. The antibody is therefore expected to have only a limited impact on receptor engagement. This is consistent with the findings of Munis *et al*. [30] showing that this 8G5F11 blocks infection after receptor binding and endocytosis. We therefore investigated whether the antibody interferes with membrane fusion using a previously described cell-cell fusion assay [34].

BSR cells expressing transiently VSV G were incubated in presence or absence of 10 nM 8G5F11 30 minutes before exposition to low pH. After incubation for 1 hour in DMEM buffered at pH 7.4, cells were fixed and analyzed (Fig 6A). As shown in Fig 6B, the presence of 8G5F11 resulted in a drastic inhibition of the fusion activity and virtually, no syncytia were observed at low pH, demonstrating that 8G5F11 efficiently blocks membrane fusion.

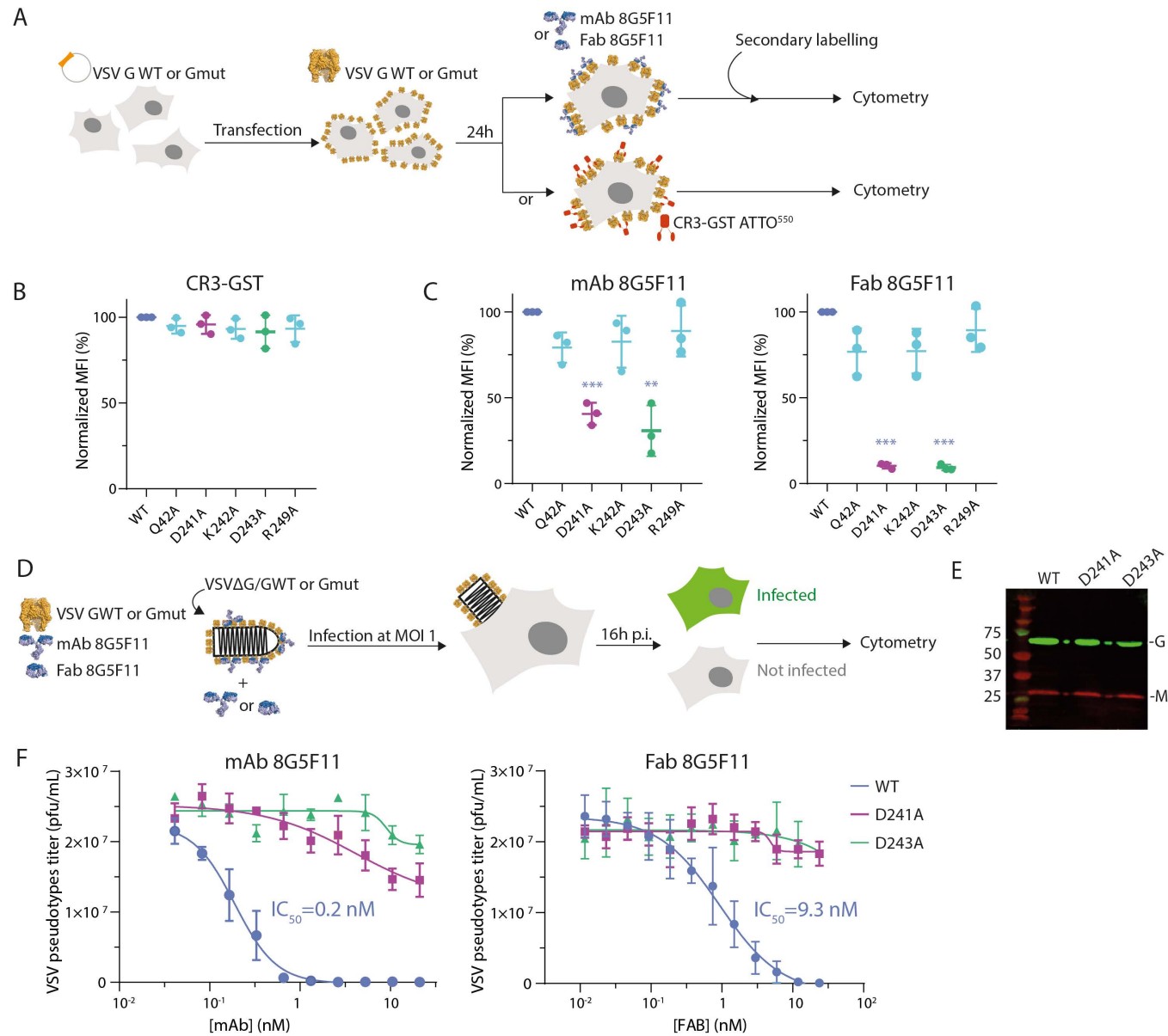

**Fig 5. Residues D241 and D243 are essential for the interaction of 8G5F11 with VSV G and for neutralization.** (A) Schematic description of the binding assays used to assess the capacity of VSV G WT and VSV G alanine mutants to bind CR domains and 8G5F11 mAb or Fab. (B) Surface expression test for VSV G WT and VSV G alanine mutants by measuring their ability to bind the conformational probe (CR3-GST labelled by and ATTO$^{550}$ fluorophore). The histogram indicates the mean fluorescence intensity of ATTO$^{550}$ positive cells for each G construct. Three independent experiments were performed. Error bars represent the standard deviation. (C) Antibody recognition assay for VSV G WT and mutants G by measuring their ability to bind 8G5F11 mAb (left panel) or Fab (right panel). Statistically significant differences with WT are indicated by stars (**$p < 0.005$, ***$p < 0.0005$). (D) Schematic description of the neutralization assay using VSVΔG/G pseudotypes. (E) Incorporation of VSV G WT and VSV G alanine mutants in VSVΔG viral particles, assessed using a polyclonal anti-VSV G and an anti-VSV M antibodies. (F) Neutralization of VSVΔG/GWT, VSVΔG/G$_{D241A}$, and VSVΔG/G$_{D243A}$ by mAb (left panel) or Fab 8G5F11 (right panel). VSV pseudotypes were preincubated with increasing concentrations of mAb or Fab. At 16 hours *p.i.,* the percentage of infected cells was determined by counting the number of cells expressing eGFP using flow cytometry. This was used to calculate the infectious viral titer. Data depict the mean with standard error for experiments performed in triplicate. Average IC$_{50}$ are indicated.

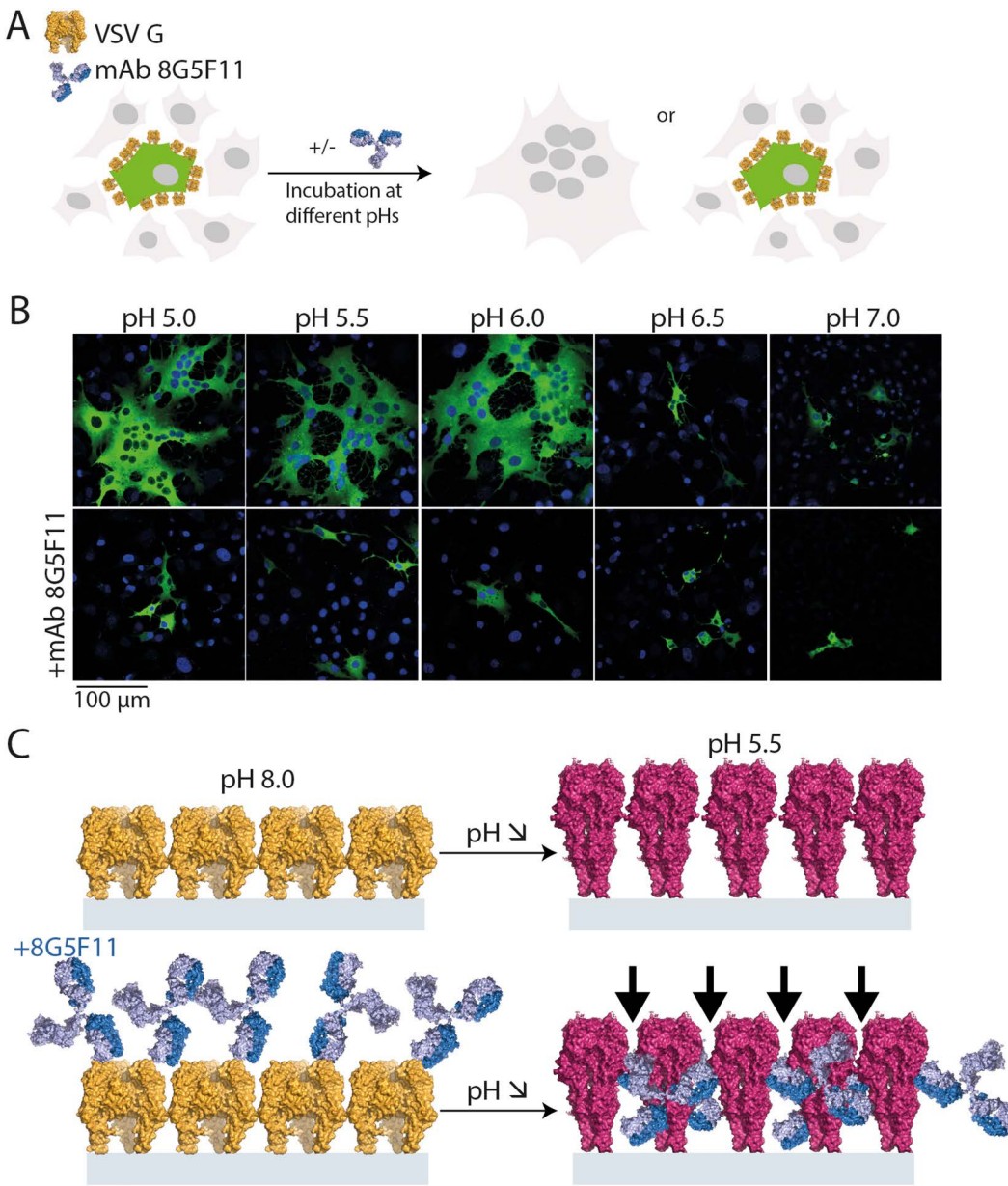

**Fig 6. Inhibition of VSV G mediated cell-cell fusion by 8G5F11.** (A) Schematic description of cell-cell fusion assay. BSR cells were co-transfected with plasmids encoding VSV G and a cytoplasmic fluorescent marker (P-GFP). Twenty-four hours later, cells were incubated in fusion buffer at different pH values in the absence or presence of 10 nM 8G5F11 and syncytium formation was assessed. (B) Cell-cell fusion assay of VSV G with or without 8G5F11 after incubation at the indicated pH values. Nuclei were stained with DAPI. All images are representative examples from three independent experiments. (C) Schematic representation of 8G5F11 mechanism of action. There are no constraints for 8G5F11 to bind its epitope on the glycoprotein in its pre-fusion conformation when G is inserted into the viral membrane. However, the density of spikes on the surface of the viral particle prevents the structural transition due to the size of the antibody and its orientation when bound to the post-fusion state, which induce significant steric hindrance (indicated by arrows).

## Discussion

In this work, we determined the structures of full-length VSV G, solubilized from viral particles, using cryo-EM SPA, both alone and in complex with the Fab derived from the neutralizing antibody 8G5F11. Overall, these structures of VSV G are largely consistent with those previously obtained by X-ray crystallography [14–16] and correspond to the pre- and post-fusion conformation of VSV G. However, the cryo-EM structures of the post-fusion state reveal new details about the organization of the CTD that were not observed in previously reported post-fusion crystalline structures of Vesiculovirus glycoproteins [16,21]. The organization of the CTD in the pre-fusion conformation was already established [15] and is confirmed here through our new cryo-EM structures. This study further demonstrate that this domain also undergoes a major conformational rearrangement upon pH decrease, involving a complete reorganization of secondary structures elements and a relocalization into a hydrophilic groove formed between two adjacent FDs. This organization of the CTD locks the post-fusion conformation and contributes to the stability of the post-fusion trimer. The correct positioning of the CTD within the hydrophilic groove is ensured by the hydrophobic patch formed by residues L423 and F425 with residue I82 of the same protomer, as well as residues L105 and P107 of the adjacent protomer. Notably, the hydrophobic character of these residues is conserved among vesiculoviruses. While our results suggest a stabilizing role for this hydrophobic patch, including F425, the presence of a tyrosine at this position in VSV New Jersey G indicates that some variation toward less hydrophobic residues may be tolerated. However, the precise contribution of this hydrophobic cluster remains to be experimentally validated.

Previous studies have shown that the low pH-induced conformational change is initiated by refolding of the CTD [15,22], which dissociates from the rest of the protomer allowing the exposure of fusion loops at the top of the molecule, thereby enabling its interaction with the target membrane. This work shows that the repositioning of the CTD, which stabilizes the post-fusion state, occurs at a late stage of the refolding process. This observation agrees with the crystal structures of intermediate conformations captured along the transition pathway of CHAV G and rabies virus glycoprotein (RABV G) [22,35].

Although early dissociation of the VSV G trimer has been previously demonstrated [8], we did not detect such intermediates in our cryo-EM analysis. All particles at both pH 8.0 and pH 5.5 appeared trimeric, with no distinct monomer observed in 2D or 3D classifications. This is consistent with the predominance of the post-fusion trimer at pH 5.5. At pH 8.0, the presence of the TMDs may contribute to trimer stabilization. In any case, monomeric intermediates are highly dynamic and thus remain difficult to capture under our experimental conditions.

The fact that we could not visualize the TMD and ID in the cryo-EM densities, is in striking contrast to the observation that we were able to visualize the part of the FD inserted into the detergent micelle. This suggests that the TMD exhibits a high degree of mobility within the micelle and probably also within the membrane. Similarly, in a cryo-EM structure of RABV G [36], solubilized using octyl glucoside that was further exchanged to A8-35 amphipol, the TMDs remained undetected despite the stabilizing effect of amphipols [37] compared with detergent. The same observation was reported for RABV G solubilized in LMNG [38]. Thus, this raises the possibility that rhabdovirus G TMDs may not associate into a stable trimeric structure. Supporting this hypothesis, tomographic reconstructions of VSV particles, showed no density potentially corresponding to TMDs [39].

This contrasts with the herpesvirus gBs for which the TMDs adopt a stable trimeric organization, which is resolved both in the pre- and post-fusion structure of HSV1 gB [40,41] and in the pre-fusion structure of HCMV gB [42]. It should be noted that the C-terminal part of the CTD corresponding to residues 432–446 for VSV G (missing in cryo-EM structures) is relatively short compared to the corresponding membrane proximal external region (MPER) of gBs. The latter is approximately 50 residues long and forms a flat-lying helical structure on the micelle, which may contribute to stabilization of TMD orientation.

It has been suggested that flexibility between the ectodomain and the TMD is required for the proper refolding of viral fusion glycoproteins [43]. Such flexibility was demonstrated for the hemagglutinin (HA) of the influenza virus [43]. However, in that case, despite the inherent flexibility of HA, the trimeric organization of TMDs could be visualized after masking the ectodomain. In contrast, applying a similar strategy for VSV G did not allow us to resolve its TMDs.

Additionally, our work provides the structures of VSV G in complex with Fab 8G5F11. VSV Indiana neutralizing mAb 8G5F11 cross-react with the glycoprotein of other Vesiculoviruses including Maraba, Alagoas, Cocal, and New Jersey viruses [30]. Our structures revealed that the epitope recognized by mAb 8G5F11 corresponds to a helix within the PHD that is exposed at the top of G protomers in the pre-fusion conformation. This region also overlaps to the major antigenic site II of RABV G [44,45]. Three G residues (D241, K242, D243) from the core of the interface between G and the paratope, among which residues D241 and D243, are essential for antibody binding and neutralization. Importantly, acidic residues in position 241 and 243 are widely conserved among Vesiculoviruses neutralized by 8G5F11, but they are absent in Piry virus G, which is not neutralized [30], as well as in Isfahan virus and CHAV Gs. It should be noted that CHAV G displays a different structural organization in this region: instead of a helix, it has a short β-hairpin [21].

Regarding the neutralization mechanism, Munis and colleagues [30] showed that the antibody most likely interferes at a post-endocytosis step. Consistent with their observations, we show (i) that the receptor binding site is distinct from the antibody recognition site, and (ii) that the antibody inhibits low pH-induced cell-cell fusion mediated by VSV G. Indeed, even though 8G5F11 allows VSV G conformational change in solution, in the context of the viral or cellular surface -where spikes are perpendicular to the membrane and likely tightly packed- the antibody's movement, in particular the shift of its binding angle from 45 to 135°, cannot be correctly accommodated, thereby preventing the structural transition (Fig 6C).

Although VSV and related viruses do not represent a major threat for human health, outbreaks in horses and livestock can cause economic losses. A deeper understanding of VSV neutralization could lead to the conception of vaccines protecting against a broad spectrum of Vesiculoviruses. Furthermore, VSV is a promising candidate for oncovirotherapy due to its ability to selectively replicate in and kill cancer cells [46–48]. However, its potential to spread uncontrollably in the bodies of patients whose immune systems have been weakened by other anticancer therapies remains a concern. In such cases, the administration of neutralizing human or humanized monoclonal antibodies could offer a potential means of control in such cases.

## Materials and methods

### Viruses and cells

BSR, a clone of BHK21 (Baby Hamster Kidney cells, ATC CCCL-10) and HEK-293T (human embryonic kidney cells expressing simian virus 40T antigen, ATCC CRL-3216) were cultured in DMEM supplemented with 10% FCS, 50 U/mL of penicillin and 50 μg/mL of streptomycin at 37°C in a humidified incubator with 5% CO2. Drosophila Schneider 2 (S2) cell line (Invitrogen) were grown in ESF 921 medium (Clinisciences) supplemented with 50 U/mL of penicillin and 50 μg/mL of streptomycin at 28°C.

VSV MS (Mudd-Summer strain, Indiana serotype), VSVeGFP (a gift from Denis Gerlier) were propagated on BSR cells. VSVΔG/G was propagated on HEK-293T cells that had been previously transfected with pCAGGS VSV G.

### Plasmid and cloning

Point mutations were introduced starting from the cloned VSV G gene (Indiana Mudd-Summer strain; Indiana serotype) in the pCAGGS plasmid. Briefly, forward and reverse primers containing the desired mutation were combined separately with one of the primers flanking the G gene to generate two PCR products. These two G gene fragments overlapped in the region containing the mutation and were assembled into pCAGGS vector, which had been linearized using EcoRI, using Gibson assembly reaction kit (New England Biolabs).

## Purification of VSV G

VSV G was directly solubilized from concentrated stocks of VSV MS using 2% w/v laurylmaltose neopentyl glycol (LMNG, Anatrace) for 22 hours at 37°C. Insoluble material was removed by centrifugation (20 minutes at 20 000g, 4°C). The supernatant was then loaded onto an anion exchange chromatography column (Resource Q 1 mL, Cytiva) equilibrated with 20 mM Tris-HCl pH 8.0, 2 mM EDTA, 0.01% w/v LMNG. VSV G was eluted with a linear gradient of NaCl (from 0 to 0.5 M NaCl). An additional purification step was performed on a size exclusion chromatography column (Superdex S200 Increase HR10/300 GL, Cytiva) equilibrated in 20 mM Tris-HCl pH 8.0, 150 mM NaCl, 2 mM EDTA, 0.005% w/v LMNG. Pure fractions containing VSV G were concentrated up to 5 mg/ml (Amicon Ultra 30 kDa cutoff, Millipore) and stored at -80°C until further use.

## Production and purification of recombinant 8G5F11 Fab

The heavy and light chains (HC and LC respectively) of 8G5F11 mAb were sequenced (Absoute antibody, https://absoluteantibody.com/) and then cloned into an insect cell expression vector derived from the pMT-puro plasmid (Addgene #17923) using Gibson Assembly (New England Biolabs), as described in [49,50]. Briefly, the LC was placed under the control of a metallothionein (MT) promoter and Drosophila BiP secretion signal (BiP ss) at the N-terminus, with a polyA tail at the C-terminus. This LC part was followed by the Fd portion of the HC, also flanked by a MT promoter and BiP ss at the N-terminus, and fused to C-terminal Strep-tag II, followed by a poly(A) tail (S5 Fig).

Pseudostable S2 cell lines expressing 8G5F11 Fab were generated by transfecting S2 cells with pMT-puro plasmid encoding 8G5F11 Fab and carrying the puromycin resistance gene. For large scale production, S2 cells were cultured in suspension at 28°C under agitation (120 rpm) in medium supplemented with 7 µg/ml puromycin, until reaching $1.5\ 10^7$ cells/ml. The cultures were then induced with 500 µM CuSO4. Five days post-induction, the supernatant was harvested and concentrated by tangential flow filtration (Vivaflow 200, VWR). Proteins were purified via their Strep-tag II using a streptavidin affinity column (StrepTrap, Cytiva) equilibrated in 20 mM Tris-HCl pH 8.0, 150 mM NaCl, 2 mM EDTA. Elution was performed in the same buffer supplemented with 3 mM α-Desthiobiotin. An additional purification step was carried out by size exclusion chromatography (Superdex 200 Increase 10/300 GL, Cytiva) equilibrated in 20 mM Tris-HCl pH 8.0, 150 mM NaCl, 2 mM EDTA. Pure fractions containing 8G5F11 Fab were concentrated (Amicon Ultra, 30 kDa cutoff, Millipore) and stored at -80 °C until further use.

## Binding assay of VSV G to mAb and Fab 8G5F11 at different pH values

8G5F11 antibody was immobilized on protein A-coated magnetic beads (Protein A Magnetic Beads, Biorad). Bead-bound antibody was incubated for 30 minutes at room temperature in buffers adjusted to different pH values ranging from 8.0 to 5.0. In parallel, purified VSV G was pre-incubated 30 minutes in the same pH-adjusted buffers. Antibody-coated beads were then mixed with the pre-incubated VSV G samples and incubated for 30 minutes at room temperature with gentle agitation. After incubation, beads were washed three times with the corresponding buffer, and bound VSV G-Fab complex was eluted by boiling in SDS sample buffer. The samples were analyzed by SDS-PAGE under reducing conditions. Interaction experiments with Fab 8G5F11 were conducted under the same conditions except that magnetic beads were coated with Strep-Tactin (MagStrep XT, IBA) instead of protein A.

## Purification of G-Fab complex

VSV G-Fab complexes were prepared by mixing VSV G with 8G5F11 Fab at a molar ratio of 1:1.2 in a total volume of 50 µL, aiming for a protein concentration of 5 mg/mL. The mix was then incubated for 30 minutes at 37 °C under gentle agitation (100–200 rpm).

Following incubation, the sample volume was adjusted to 1 mL with buffer C (10 mM Tris-HCl pH 8.0, 100 mM NaCl, 2 mM EDTA, 0.005% w/v LMNG) and loaded onto size exclusion chromatography (Superdex 200 Increase 10/300 GL, Cytiva) equilibrated in the same buffer. Fractions corresponding to the G-Fab complex were pooled and concentrated (Amicon Ultra, 30 kDa cutoff, Millipore) before downstream use.

## Preparation of cryo-EM grids and data collection

For VSV G "alone" samples, grids were prepared by applying 4 µL of VSV G concentrated at 1.95 mg/ml to negatively glow discharged (25 mA, 45 seconds) UltraFoil Au grids 300 R1.2/1.3 (QUANTIFOIL) prior to plunge freezing using a Vitrobot Mark IV (FEI) (20°C, 100% humidity, 5.5 seconds blot time and 0 blot force). For VSV G in the post-fusion conformation, MES 1 M pH 5.5 was added to VSV G to reach final concentration of MES of 100 mM in the sample 30 minutes at 37°C just before freezing. Grids were pre-screened on a 200 kV Glacios and automated data collection was performed at the beamline CM01 of ESRF (Grenoble, France) on a 300 kV Titan Krios equipped with a K3 direct electron detector, coupled to an energy filter (Bioquantum LS/967, Gatan Inc, USA) [51]. 8,840 and 6,864 movies were recorded with EPU (Thermofisher) for the pre- and post-fusion of VSV G respectively, with a pixel size of 1.05 Å/pixels, a total dose of 40.4 e-/Å and a defocus range of -0.8 to -2.2 µm.

Cryo-EM grids of VSV G in complex with 8G5F11 Fab at pH 8.0 were prepared by applying 4 µL of the complex at a concentration of 0.8 mg/mL to negatively glow discharged (25 mA, 45 seconds) UltraFoil Au grids 300 R1.2/1.3 (QUANTIFOIL) prior to plunge freezing using a Vitrobot Mark IV (FEI) (30°C, 100% humidity, 5.5 seconds blot time and blot force 0). SerialEM based automated data collection was performed on a 200 kV Glacios electron microscope equipped with a K2 summit direct electron detector. 15947 movies were recorded with EPU (ThermoFisher) with pixel size of 1.145 Å/pixels, a total dose of 38.5 e-/Å and a defocus range of -0.8 to -2.2 µm.

MES 1 M pH 5.5 was added to G-Fab complex to reach a final concentration of MES of 100 mM in the sample 30 minutes at 37°C just before freezing. Cryo-EM grids of G-Fab complex at pH 5.5 were then prepared by applying 3.5 µL of the complex at 0.8 mg/mL to negatively glow discharged (25 mA, 45 seconds) UltraFoil Au grids 300 R1.2/1.3 (QUANTIFOIL) prior to plunge freezing using a Vitrobot Mark IV (FEI) (30°C, 100% humidity, 5.5 seconds blot time and blot force 0). The grids were pre-screened on a 200 kV Glacios electron microscope at IBS (Grenoble, France) and automated data collection was performed at the beamline CM01 of ESRF (Grenoble, France) on a 300 kV Titan Krios equipped with a K3 direct electron detector, coupled to an energy filter (Bioquantum LS/967, Gatan Inc, USA) [51]. 19423 movies were recorded with EPU (Thermofisher) with pixel size of 0.84 Å/pixels, a total dose of 49.71 e-/Å and a defocus range of -0.8 to -2.2 µm.

## Cryo-EM data processing and model building

Image processing was performed using cryoSPARC software [52] and is summarized in S2-S3 and S6-S7 Figs. Recorded movies were dose-weighted and motion-corrected using cryoSPARC Patch Motion Correction program. CTF parameters were determined using cryoSPARC Patch CTF program. Particles were at first automatically picked using circular and elliptical blobs with a diameter between 100 and 230 Å for VSV G pH 5.5, between 100 Å and 250 Å for VSV G pH 8.0, between 150 Å and 300 Å for VSV G-Fab pH 8.0 and 80 Å and 300 Å for VSV G-Fab pH 5.5. A first round of particles picking was performed for each data set to select correct side views 2D classes. Topaz neural network was then used to enrich particles data set in particles side views and visible micelle [53]. Several rounds of 2D classification were then performed in order to clean particle data sets. *Ab-initio* reconstruction was used to generate at least 2 models to classify particles among them. Several rounds of Homogeneous refinement and Non-uniform refinement [54] were performed without imposed symmetry. The best models were selected for global and local (per-particle) CTF refinement to improve particle, map quality and resolution. C3 symmetry was only imposed on VSV pH 8.0 and VSV-Fab pH 5.5 reconstructions. Reported maps resolutions were estimated by gold-standard Fourier shell correlation (FSC) of 0.143 criterion.

Model building for protein chains was performed using COOT v.0.9.6 EL [55]. Model refinement and validation was performed using Phenix v.1.20.1 [56,57] and ISOLDE [58]. Fab model building was based on an initial model generated from AlphaFold2 [59] then fitted into the cryo-EM map and manually refined in COOT v.0.9.6 EL [55]. Geometry was validated using MolProbity [60].

## Biolayer interferometry analysis (BLI)

BLI experiments were performed using an Octet RED96e (Sartorius, Fremont, CA, USA) instrument operated at 20°C under 1000 rpm stirring with Octet BLI Discovery software. The 8G5F11 mAb was captured on Protein A-functionalized sensors at 20 μg/mL during 40 seconds in acetate 10 mM pH 4.0. Fab 8G5F11 were covalently immobilized with random orientation on Amine Reactive 2nd Generation sensors (AR2G) at 20 μg/mL during 600 seconds in acetate 10 mM pH 4.0 using standard EDC-NHs chemistry. Interactions with VSV $G_{ecto}$, from 6.25 to 150 nM in running buffer (Tris-HCl 20 mM pH 8.0, NaCl 100 mM, EDTA 2 mM), were monitored during 300 seconds (association phase). Dissociation step was set at 600 seconds on the mAb surfaces and 1200 seconds on the Fab surfaces respectively. A bare sensor was always added as a reference surface to control the non-specific binding.

Data analysis was performed on Octet Analysis Studio software and kinetics constant (ka and kdis) were extracted by fitting the data with a 1:1 binding model and equilibrium dissociation constant KD was calculated.

## Binding assay of 8G5F11 mAb and Fab to HEK-293T expressing VSV G WT or VSV G mutants at their surface

CR3-GST was produced in *E. Coli* (C41(DE3); Lucigen) and labeled with the fluorescent dye ATTO$^{550}$ NHS ester (Sigma Aldrich) following the instruction of the manufacturer, as previously described (Nikolic et al., 2018). The labelled probe was then diluted at a concentration of 50 mM and stored at -80°C for further use. The labelling ratio was estimated to be around 2 dyes per CR3-GST molecules.

Cell surface expression levels of G WT or mutant G were estimated by adding GST-CR3 ATTO$^{550}$ at a concentration of 500 μM on transfected cells 24 hours post-transfection. The fluorescence of cells was measured using Beckman CytoFlex S cytometer.

For binding assays, HEK-293T cells were transfected with pCAGGS plasmids encoding G WT or G mutants using the calcium phosphate transfection method. Twenty-four hours after transfection, cells were collected and incubated with either 8G5F11 mAb (at a concentration of 1.3 $10^{-3}$ mM) or 8G5F11 Fab (at a concentration of 2.5 $10^{-5}$ mM) for 45 minutes at RT. Goat anti-mouse Alexa fluor 488 (at 1.3 $10^{-3}$ mM) was added to the cells to accessed the binding of mAb 8G5F11. Anti-strep-tag coupled to PE fluorophore was added to the cells to accessed the binding of Fab 8G5F11. The fluorescence of cells was measured using a Beckman CytoFlex S cytometer.

## VSVeGFP neutralization assay

HEK-293T cells at 70% confluence in 96-well plates were infected at MOI 1 with VSVeGFP pre-incubated for 30 minutes with increasing concentrations of 8G5F11 mAb or Fab. The percentage of infected cells (eGFP-positive) was determined using a Beckman CytoFlex S cytometer, 5 hours *p.i.*. Experimental MOIs were determined using the equation MOI=− ln[p(0)], where MOI is the multiplicity of infection and p(0) is the proportion of non-infected cells. Viral titer was deduced from the MOI, considering the volume and dilution of the viral inoculum and the total number of cells.

## VSVΔG-GFP pseudotypes production

HEK-293T cells at 70% confluence were transfected with pCAGGS plasmids encoding G WT or mutant G using the calcium phosphate transfection method. Twenty-four hours after transfection, cells were infected with VSVΔG pseudotyped by VSV G WT at MOI 1. Two hours *p.i.*, cells were washed to remove the inoculum. Then, at 16 hours *p.i.*, supernatants

containing the pseudotypes were collected. The infectious titers of the pseudotyped viruses was determined by counting cells expressing the GFP using a Beckman CytoFlex S cytometer 16 hours *p.i.*. Experimental MOIs were determined using the equation MOI=−ln[p(0)], where MOI is the multiplicity of infection and p(0) is the proportion of non-infected cells. Viral titer was deduced from the MOI, considering the volume and dilution of the viral inoculum and the total number of cells.

### VSVΔG pseudotypes neutralization assay

HEK-293T cells at 70% confluence in 96-well plates were infected at MOI 1 with VSVΔG pseudotypes pre-incubated for 30 minutes with increasing concentrations of 8G5F11 mAb or Fab. The percentage of infected cells (GFP-positive) was determined using a Beckman CytoFlex S cytometer, 16 hours *p.i.*. Experimental MOIs were determined using the equation MOI=−ln[p(0)], where MOI is the multiplicity of infection and p(0) is the proportion of non-infected cells. Viral titer was deduced from the MOI, considering the volume and dilution of the viral inoculum and the total number of cells.

### Western blot analysis

G WT and G mutants incorporation into the membrane of the pseudotyped VSVΔG particles was assessed, after concentrating viral supernatants, by Western blot analysis using rabbit-polyclonal anti-VSV G antibody (inhouse) (1/5000) and a mouse-monoclonal anti-VSV M antibody (1/5000). Goat anti-rabbit DyLight 800 conjugate and Goat anti-mouse DyLigth 680 conjugate were used as secondary antibodies. Blots were imaged using a LI-COR Biosciences Odyssey.

### Cell-cell fusion assay

Cell-cell fusion was assessed as previously described [2,22]. Briefly, BSR cells were plated in 6 well plates (~70% confluence) and co-transfected with pCAGGS encoding VSV G and a P-GFP plasmid (Rabies virus phosphoprotein fused to GFP that diffuse in the cytoplasm). At 24 hours post-transfection, cells were incubated for 10 minutes at 37°C with 0.5 mL of DMEM containing 10 nM 8G5F11 mAb, followed by exposure to fusion buffer (DMEM supplemented with 10 mM MES) adjusted to the indicated pH (5.0-7.0). After a single wash, cells were incubated for 1 hour at 37°C in DMEM containing 10 mM HEPES pH 7.4 and 1% BSA. Cells were then fixed with 4% paraformaldehyde in PBS for 15 minutes, nuclei stained with DAPI, and syncytium formation was examined using a Leica SP8 confocal microscope (40× objective).

### Statistical analysis

All numerical data were calculated and plotted with mean ±SD resulting from three independent biological replicates. Results were analyzed using Prism v8.0.1 (GraphPad Software).

For the recognition assay, results are presented as normalized MFI. To do so, the geometric mean of the measured MFI for each condition of each experimental replicate was calculated. The geometric means were then normalized to the one of the WT. Results were compared using T-tests to compare means of biological replicates. Experiments were performed in triplicates.

For neutralization and pseudotypes assays, experimental titers were determined based on experimental MOIs. Dose-response parameters, namely Hillslope and $IC_{50}$, were determined using a log(inhibitor) vs response - variable slope model.

### Supporting information

**S1 Fig. Purification of VSV G.** Elution profile of VSV G on a Superdex S200 increase HR 10/300 GL (Cytiva) and Coomassie-stained SDS-PAGE analysis of VSV G purification steps (1: concentrated VSV preparation, 2: VSV G after anion exchange chromatography step, 3: VSV G after size exclusion chromatography step).
(TIF)

**S2 Fig. Cryo-EM data processing pipeline for VSV G at pH 8.0.** (A) Representative electron micrograph. (B) Cryo-EM data processing workflow in cryoSPARC. (C) Final density used for model building (left) and local resolution map calculated and plotted onto the sharpened VSV G reconstruction.
(TIF)

**S3 Fig. Cryo-EM data processing pipeline for VSV G at pH 5.5.** (A) Representative electron micrograph. (B) Cryo-EM data processing workflow in cryoSPARC. (C) Final density used for model building (left) and local resolution map calculated and plotted onto the sharpened VSV G reconstruction.
(TIF)

**S4 Fig. Representation of the interaction between the CTD and the FD by 2D plots (calculated with *Dimplot*).** (A) Representation of the interaction of the CTD and the FD at pH 5.5. The left panel shows *Dimplot* of VSV G CTD (residues 419–426, in magenta) with the FD from protomer A (in yellow), and the right panel shows the *Dimplot* of the CTD with the FD from protomer B (in green). (B) *Dimplot* of VSV G CTD with FD at pH 8.0. Putative hydrophobic interactions are depicted in red dashed lines, and polar interactions are in light blue.
(TIF)

**S5 Fig. Sequence alignment of Vesiculovirus glycoproteins.** Conserved residues are highlighted in dark blue boxes, while similar residues are shown in lighter blue. VSVI G domains are indicated above the sequence and depicted according to Fig 1A color code. Asparagine carrying N-glycosylation are marked with an orange circle. The residues constituting the fusion loops are framed in yellow boxes. Residues belonging to 8G5F11 epitope are framed in cyan boxes. The residues constituting the hydrophobic patch stabilizing the CTD in the post-fusion conformation are indicated by brown arrows.
(TIF)

**S6 Fig. Purification of VSV G-Fab complex for cryo-EM studies.** (A) Diagram of the construction used to produce 8G5F11 Fab. (BIP ss = BIP signal sequence; LC = light chain; Fd = part of the heavy chain composing the Fab; StrepII = Strep-tag II). (B) Elution profile of Fab on a Superdex S200 HR 10/300 (Cytiva) in 20 mM Tris-HCl pH 8.0, 150 mM NaCl, 2 mM EDTA (left panel) and Coomassie-stained SDS-PAGE analysis of purified Fab (right panel). (C) Schematic description of VSV G-Fab complex assembly for cryo-EM studies. (D) Elution profile of VSV G-Fab complex on a Superdex S200 HR 10/300 (Cytiva) in 20 mM Tris-HCl pH 8.0, 150 mM NaCl 2 mM EDTA (left panel) and Coomassie-stained SDS-PAGE analysis of purified VSV G-Fab.
(TIF)

**S7 Fig. Cryo-EM data processing pipeline for VSV G at pH 8.0 in complex with Fab.** (A) Representative cryo-electron micrograph. (B) Cryo-EM data processing workflow in cryoSPARC. (C) Final density used for model building (left panel) and local resolution map calculated and plotted onto the sharpened VSV G reconstruction (middle panel) and representative fit of atomic model of G and Fab into density (right panel).
(TIF)

**S8 Fig. Cryo-EM data processing pipeline for VSV G at pH 5.5 in complex with Fab.** (A) Representative cryo-electron micrograph.(B) Cryo-EM data processing workflow in cryoSPARC. (C) Final density used for model building (left panel) and local resolution map calculated and plotted onto the sharpened VSV G reconstruction (upper right panel) and representative fit of atomic model of G and Fab into density (lower right panel).
(TIF)

**S9 Fig. 8G5F11 and LDL-R binding sites on VSV G.** Binding footprint of 8G5F11 and CR2 (A) or CR3 (B) on VSV G.
(TIF)

**S1 Table. Model and map statistics.**
(DOCX)

**S2 Table. Average RMSD between aligned Cα (in Å) of the overall structure and different regions.**
(DOCX)

**S3 Table. Table of the KDs, association (Ka) and dissociation rate constants (Kdis) for $G_{ect}$ and 8G5F11 mAb and Fab.**
(DOCX)

## Acknowledgments

We acknowledge the cryoEM and PIM platforms of I2BC supported by the French Infrastructure for Integrated Structural Biology. We acknowledge the European Synchrotron Radiation Facility (ESRF) for provision of beamtime on CM01 (proposal number MX-2440 and MX-2557), and we thank A. Grinzato and L. McGregor for assistance in using the beamline. We thank I. Duquenois for technical assistance with the cytometry experiments. We also thank C. Lagaudrière-Gesber for careful reading of the manuscript.

## Author contributions

**Conceptualization:** Yves Gaudin, Aurelie A ALBERTINI.

**Data curation:** Marie Minoves, Malika Ouldali, Eleftherios Zarkadas, Guy Schoehn, Aurelie A ALBERTINI.

**Formal analysis:** Marie Minoves, Malika Ouldali, Laura Belot, Magali Noiray, Eleftherios Zarkadas, Guy Schoehn, Yves Gaudin, Aurelie A ALBERTINI.

**Funding acquisition:** Guy Schoehn, Yves Gaudin, Aurelie ALBERTINI.

**Investigation:** Laura Belot, Sarah Johari, Yves Gaudin, Aurelie A ALBERTINI.

**Methodology:** Marie Minoves, Eleftherios Zarkadas, Guy Schoehn.

**Project administration:** Aurelie A ALBERTINI.

**Supervision:** Aurelie A ALBERTINI.

**Validation:** Malika Ouldali, Stéphane Roche, Eleftherios Zarkadas, Guy Schoehn, Yves Gaudin, Aurelie A ALBERTINI.

**Visualization:** Marie Minoves, Malika Ouldali, Stéphane Roche, Aurelie A ALBERTINI.

**Writing – original draft:** Yves Gaudin, Aurelie A ALBERTINI.

**Writing – review & editing:** Marie Minoves, Laura Belot, Stéphane Roche, Guy Schoehn.

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
