## [Decision Letter · Decision Letter 0]

6 Jun 2025

PPATHOGENS-D-25-00837

Structures of vesicular stomatitis virus glycoprotein G alone and bond to a neutralizing antibody

PLOS Pathogens

Dear Dr. ALBERTINI,

Thank you for submitting your manuscript to PLOS Pathogens. After careful consideration, we feel that it has merit but does not fully meet PLOS Pathogens's publication criteria as it currently stands. Therefore, we invite you to submit a revised version of the manuscript that addresses the points raised during the review process.

I urge the authors to seriously consider additional experiments to understand further the neutralization mechanisms of 8G5F11 since the pre and post-fusion structures of VSV-G have been previously published (although the additional visualization of the CTD region is new, it alone does not justify a new publication in this journal).   

Please submit your revised manuscript within 30 days Aug 05 2025 11:59PM. If you will need more time than this to complete your revisions, please reply to this message or contact the journal office at plospathogens@plos.org. Please include the following items when submitting your revised manuscript:

We look forward to receiving your revised manuscript.

Kind regards,

Benhur Lee

Section Editor

PLOS Pathogens

Benhur Lee

Section Editor

PLOS Pathogens

Sumita Bhaduri-McIntosh

Editor-in-Chief

PLOS Pathogens

orcid.org/0000-0003-2946-9497

Michael Malim

Editor-in-Chief

PLOS Pathogens

orcid.org/0000-0002-7699-2064

**Journal Requirements:**

1) Please provide an Author Summary. This should appear in your manuscript between the Abstract (if applicable) and the Introduction, and should be 150-200 words long. The aim should be to make your findings accessible to a wide audience that includes both scientists and non-scientists. Sample summaries can be found on our website under Submission Guidelines:

https://journals.plos.org/plospathogens/s/submission-guidelines#loc-parts-of-a-submission

- ® on pages: 20, 21, 22, and 25.

3) Please ensure that the funders and grant numbers match between the Financial Disclosure field and the Funding Information tab in your submission form. Note that the funders must be provided in the same order in both places as well.

**Reviewers' Comments:**

Reviewer's Responses to Questions

**Part I - Summary**

Reviewer #1: In their manuscript, Minoves et al. present four cryo-EM structures of vesicular stomatitis virus glycoprotein, VSV-G: Two in pre-fusion conformation, including one bound by Fab 8G5F11; two in post-fusion conformation, including one in complex with the same Fab. The transmembrane and intraviral regions, however, are not resolved due to their inherent flexibility. Therefore, the structures presented correspond essentially to the ectodomain of the glycoprotein.

The new structures provide marginally new information, as both the VSV-G pre- and post-fusion ectodomain structures have been previously determined using X-ray crystallography by the same research group. While the new cryo-EM derived pre-fusion structure did not yield new structural insights, the new post-fusion structure resolves an additional 16 residues in the CTD region that was previously not observed in the crystal structure. Direct comparison of this region between the two conformations demonstrates the rearrangement it undergoes during pH-induced structural changes. The authors pinpointed a number of hydrophobic interactions in this region that likely contribute to CTD stability, but these are not validated experimentally. The authors also provided no further commentary regarding the specifics of the other interactions (eg. hydrogen bonds and salt bridges) governing this region.

Additionally, the Fab-G structures revealed the binding epitopes of a broadly neutralising mAb 8G5F11. While the critical epitope residues (Asp-Lys-Asp) have been previously mapped by Munis et al., 2018 (PMID: 30232190), the structures in this manuscript verified their contributions in molecular details, coupled with experimental validation using pseudotyped VSV with mutant Gs. These important insights will contribute to the development of mAb-based prophylactics and vaccines for livestock and human. This work, however, have not investigated the mechanism underlying VSV neutralisation by 8G5F11. The lack of experimental study on 8G5F11 mechanism of action reduces the overall impact of the manuscript.

Overall, the manuscript is well written. The presented data supports the conclusions and will be relevant and of interest for researchers working on vesiculoviruses and class III viral fusion proteins.

Reviewer #2: In the study, the group of Aurélie A. Albertini and Yves Gaudin have determined the structures of full-length VSV-G, solubilized from viral particles, both alone and in complex with the neutralizing antibody 8G5F11. This study is significant because the cryo-EM structures of the post-fusion VSV-G provide new structural details about the organization of the CTD, which were absent in previously reported post-fusion crystal structures of Vesiculovirus Gs. The structure demonstrates that in the post-fusion state of VSV-G, the CTD interacts with adjacent protomer, suggesting a potential role in stabilizing the homo-trimeric assembly. Moreover, these structural insights provide valuable guidance for investigations of rabies virus G (RABV-G), as the corresponding CTD in RABV-G remains unresolved to date.

Overall, this work makes substantial contributions to rhabdovirus G protein studies. However, revisions are necessary to address unclear descriptions and typographical errors in the Results, Methods, and Figures to improve manuscript quality.

**Part II – Major Issues: Key Experiments Required for Acceptance**

Reviewer #1: The following experiments would add strength to the hypotheses made by the authors:

1. Lines 150-152: In addition to the pinpointed hydrophobic interactions, highlighting other interactions such as hydrogen bonds and salt bridges (given the prominence of the hydrophilic groove), contributed by the newly resolved CTD region, would allow direct comparison of the bonds required to break and/or formed at the equivalent region during pH-induced structural changes from pre- to post-fusion states. This could be done with detailed supplementary close-up diagrams for the interactions formed at this region or a detailed 2D interaction schematic.

2. Lines 391-397: The authors hypothesized a few possible neutralisation mechanisms for mAb 8G5F11, including interfering with VSV-G conformational changes. This should be validated experimentally. As the receptor for VSV, LDL-R, only binds pre-fusion G and does not compete with 8G5F11 binding, the authors could use the receptor (eg. CR3 domain) as conformational probe in a BLI real-time binding experiment to test its binding to Fab-G complexes buffer-exchanged from neutral to acidic pH, and vice versa. Alternatively, the authors could also generate cryoEM 2D classes to observe the effect of pH change on Fab-bound G conformation, if any.

As Munis et al., 2018 (PMID: 30232190) showed that 8G5F11 blocks infection after receptor binding and endocytosis but before genome reverse transcription, the authors could use a membrane fusion assay or a cell-cell fusion assay to test if the mAb interferes with membrane fusion at acidic endosomal pH. The data would further support the BLI/2D averaging experiments above.

Interestingly, in Figure S5 (C), the authors indicated that the G-Fab complex was formed at pH 8.0, followed by acidification to pH 5.5 to obtain the post-fusion G-Fab structure. Does this mean that 8G5F11 does not hinder conformational changes from pre- to post-fusion? This needs to be clarified.

Reviewer #2: None.

**Part III – Minor Issues: Editorial and Data Presentation Modifications**

Reviewer #1: The following clarification and/or additional data would improve the quality of the manuscript:

1. The antibody name 8G5F11 should be mentioned in the Abstract.

2. Lines 113-114: Given that pre-to-post structural rearrangement has been proposed to involve transient G dissociation (PMID: 22383886), did the authors see any evidence for de-trimerisation of the VSV-G trimer upon acidification, eg. in micrographs, or in the 2D or 3D classes? A comment on this in the Discussion section would be helpful for readers to understand.

3. Lines 133-136 & 364: It would be of interest to readers if the authors could comment on the angular degree of freedom for the TMD flexibility, for instance, from an estimation using the 2D class averages or masked 3D classification, if possible. This will allow readers an appreciation for the high structural dynamics of the region, while strengthening authors' comparison to the more rigid HCMV and HSV-1 gB, of which the TMDs were resolved structurally.

4. Line 154-155, 354-355: VSV NJ G has a polar Y425 instead of nonpolar F425 at the identified CTD hydrophobic cluster. Does this mean this site is dispensable, or compensated by site 423, or the aromatic nature of the side chain is more important than the charge? It's unclear from Figure 2C if it was the side chain or main chain of F425 that interacts with I82. Given that the importance or contribution of this hydrophobic patch was not tested, the authors should mention caveat for their claim.

5. Lines 378-379: The RABV-G structure determined in reference [38], Ng et al., was solubilised using bOG detergent and subsequently exchanged into amphipol. The authors should correct this, indicating that solubilisation using amphipol, an amphipathic polymer commonly less denaturing and more rigid than detergent, similarly demonstrates the flexibility of TMD regions. Authors should also cite PMID: 35714192 as additional reference which solubilised RABV-G using LMNG.

6. Lines 392-393: Supporting reference(s) should be provided for this hypothesis to support the rationale of this mechanism.

7. Lines 491-492: The authors should explain the choice to impose C3 symmetry only on VSV pH8.0 and VSV/Fab pH5.0 reconstructions but not VSV pH5.0 and VSV/Fab pH8.0? Are there any interesting asymmetric features among the three protomers of the molecule in the C1 structures?

8. Minor comments regarding figures:

Fig 1A: The authors should indicate in the legend that the two pinned icons above the VSV-G domain organisational diagram are N-linked glycosylations.

Fig S2, S3, S6, S7: ‘Particles’ spelling mistake in the workflow diagram.

Fig S5B & S5D: Non-sensible ladder molecular weight from 25 to 50 back to 37 to 50 to 75.

Fig S7: Representative fit of atomic model of G and FAb into density is missing in panel C.

Reviewer #2: 1, Figure 1A contains upper and lower panels. For clarity, the figure legend should explicitly define the color representations in both schematics. Additionally, the upper panel of Figure 1A reveals two glycosylation chains. It would be helpful to label the positions of the two glycosylation sites (N163 and N325) in the schematic diagram. Correspondingly, in Figure1B and 1C, the locations of N163 and N325 glycosylation should be labeled in both the pre-fusion and post-fusion structures. This will help readers to understanding the conformational changes of the VSV-G protein.

2, In Figure 2C, the left panel displays the side chains of L423, F424, F425 from the CTD and I82 from the FD in protomer A, and L105 in protomer B. The right panel shows the side chains of L423, F425 from the CTD and I82 from the FD in protomer A, and L105 and P107 in protomer B. For consistency, the same amino acids should be selected to display their side chains in the two panels. The figure legend should also explain why these residues are selected.

3, Figures 2C, 2E, 4E, and 4F employ yellow and purple dashed lines with varying thicknesses to represent molecular interactions. The authors should clarify in the figure legends what types of molecular interactions these differently dash lines represent. Alternatively, revise the figures to present more clearly the interaction details.

4, The authors should remove the shadow effects from Figures 2C, 2E, 4E and 4F to improve image readability.

5, The BLI assay results in Figures 3B and 3D should include affinity values, do not only provide them in Table S3. In addition, in Figure 4D, the fitted curves show poor agreement with the actual data at 150 nM and 100 nM concentrations, which may affect affinity calculation accuracy.

6, In Figure 3, 'mAb' should be replaced by 'Full-length antibody' to match the 'Fab' nomenclature better.

7, In Lines 540-549, the 'VSVΔG-GFP pseudotypes neutralization assay' section only describes pseudotyped virus preparation and titration, lacking the method of neutralization assay.

8, Line 289. The heading 'Characterization of 8G5F11 binding site' would be revised as 'Characterization of the key residues for 8G5F11 binding'.

9, There appears to be a discrepancy in pH values reported for the post-fusion structures: Table S1 lists two structures as "VSV Gfl pH 5.0" and "VSV G-Fab pH 5.0", while the main text (e.g., lines 74 and 78) describes the VSV G post-fusion structure as being determined at pH 5.5.

10, In the Materials and Methods section, some decimal points are incorrectly formatted as commas. For instance, in line 468. "0,84 Å/pixels" and "49,71 e-/Å") should be revised to “0.84 Å/pixels, 49.71 e-/Å”.

PLOS authors have the option to publish the peer review history of their article (what does this mean? ). If published, this will include your full peer review and any attached files.

**Do you want your identity to be public for this peer review?** For information about this choice, including consent withdrawal, please see our Privacy Policy .

Reviewer #1: No

Reviewer #2: No

**Figure resubmission:**
---

## [Editor Report · Decision Letter 1]

29 Sep 2025

Dear Dr ALBERTINI,

We are pleased to inform you that your manuscript 'Structures of vesicular stomatitis virus glycoprotein G alone and bond to a neutralizing antibody' has been provisionally accepted for publication in PLOS Pathogens.

Best regards,

Benhur Lee

Section Editor

PLOS Pathogens

Benhur Lee

Section Editor

PLOS Pathogens

Sumita Bhaduri-McIntosh

Editor-in-Chief

PLOS Pathogens

orcid.org/0000-0003-2946-9497

Michael Malim

Editor-in-Chief

PLOS Pathogens

orcid.org/0000-0002-7699-2064